# Weight perceptions in older adults: findings from the English Longitudinal Study of Ageing

Sarah E Jackson [ORCID],[1] Lee Smith,[2] Andrew Steptoe[1]

¹Behavioural Science and Health, University College London, London, UK
²Department of Life Sciences, Anglia Ruskin University–Cambridge Campus, Cambridge, UK

**Correspondence to**
Dr Sarah E Jackson;
s.e.jackson@ucl.ac.uk

## ABSTRACT

**Objectives** To explore weight perceptions in a large, nationally representative sample of older adults, and the extent to which they differ according to age and perceived health status.

**Setting** England.

**Participants** 5240 men and women (≥50 years old) participating in the English Longitudinal Study of Ageing (2016/2017).

**Main outcome measures** Weight perception was self-reported as too heavy, too light or about right.

**Results** The majority of older adults endorsed a weight perception that matched their (objectively measured) body mass index (BMI) classification. However, 1 in 10 (9.9%) older adults classified by BMI as normal weight (18.5–24.9 kg/m²) felt too light, with women at the upper end of the older age spectrum (OR=1.04, 95% CI 1.01 to 1.09), and men (OR=3.70, 95% CI 1.88 to 7.28) and women (OR=2.61, 95% CI 1.27 to 5.35) in poorer health more likely to do so. Almost half (44.8%) of older adults classified as overweight (25–29.9 kg/m²) and 1 in 10 (10.3%) classified as obese (≥30 kg/m²) felt about the right weight, with this observed more frequently among men and women at the upper end of the older age spectrum (OR range 1.04–1.06).

**Conclusion** Older adults' perceptions of their own weight generally correspond with traditional BMI cut-offs for normal weight, overweight and obesity. However, a substantial minority 'underestimate' their weight status, with those at the upper end of the age spectrum and those in poorer health more likely to do so.

**Strengths and limitations of this study**

► Data were from a large, representative sample of older adults in England.
► Height and weight were objectively measured for calculation of body mass index.
► However, there was a substantial amount of missing data, so findings may not generalise to the entire population.
► If those who were more concerned about their weight were more likely to decline to be measured, our results may underestimate the proportion of older adults across all weight groups who consider themselves to be too heavy.

Understanding how older adults perceive their own weight status, and the extent to which this is influenced by their age and health status, is important for informing targeted recommendations and interventions to promote healthy weight in later life.

Older adults' weight perceptions may be influenced by several social and physiological factors. On a societal level, there are strong preferences for slimness in women and lean muscularity in men.[5] Some evidence suggests that older people may be less influenced by, and feel less pressure to attain, cultural body weight ideals.[6 7] Nonetheless, body dissatisfaction is evident in mid-life and later life[8–12] and may even increase as age-related changes in body composition widen the discrepancy between ideal and actual body image.[13] Ageing is associated with an increase in fat mass, loss of lean muscle mass and redistribution of adipose tissue to the abdominal region[14]; changes that may occur without concomitant changes in body weight and BMI.[15]

Few previous studies have examined older adults' weight perceptions. Those that have have indicated age-related changes in perceptions of body weight in relation to actual BMI category. For example, in a sample of adult Korean women (n=8906, ages 20–79 years),

## INTRODUCTION

The global obesity epidemic and ageing population are major public health concerns. With excess weight becoming increasingly 'normal', public perceptions of what constitutes a healthy body weight have become more inaccurate over time, with increasing numbers perceiving a body mass index (BMI) in the overweight or obese range (≥25 kg/m²) to be 'about right'.[1] The numbers of older adults with overweight (BMI 25–29.9 kg/m²) and obesity (BMI ≥30 kg/m²) are rising rapidly, due to concurrent increases in the number of adults who reach older age and the proportion who carry excess weight.[2–4]

older women were more likely than younger women to underestimate their weight status relative to their actual BMI category (50.7% of 70–79 year-olds vs 12.6% of 20–29 year-olds) and less likely to overestimate their weight status (7.4% vs 18.7%, respectively).[16] Similarly, in a study of older Dutch men and women (n=1295, ages 60–96 years), the proportion of women who underestimated their weight increased with age (OR=2.97, 95% CI 1.59 to 5.57 for 80–96 year-olds vs 60–69 year-olds), although no such pattern was found for men.[17] However, it was not clear from these studies whether these differences are driven by differences between older and younger people, for example, relating to health status. Qualitative research suggests at least some older adults believe carrying extra weight could be protective in times of illness,[12] which may mean older people's weight perceptions are influenced by current perceptions of health status or future health concerns.

The present study therefore aimed to explore perceptions of weight in a large, nationally representative sample of older adults living in England, and the extent to which they differ according to age and perceived health status.

## METHODS
### Study population
Data were from the English Longitudinal Study of Ageing (ELSA), a panel study of men and women aged ≥50 years living in England. Full details of the study's sampling procedure and methods are available elsewhere.[18] The present analyses use data collected in the eighth wave of ELSA (collected 2016/2017), as this is the only wave in which weight perceptions have been assessed. Of the 8445 participants interviewed, 5240 (62.0%) had complete data on all variables of interest and comprised our analytic sample.

### Measures
#### Weight perception
Weight perception was assessed with the question, '*Given your age and height, would you say that you are about the right weight, too heavy, or too light?*'

#### Anthropometric data
Weight was measured by nurses to the nearest 0.1 kg using portable electronic scales. Height was measured in Wave 6 (it was not included in the Wave 8 assessment) to the nearest millimetre using a portable stadiometer. Nurses recorded any factors that might have compromised the reliability of the measurements (eg, participant was stooped/unwilling to remove shoes) and these cases were excluded. BMI was calculated as weight in kilograms divided by the square of height in metres, and categorised as underweight ($<18.5 \, kg/m^2$), normal weight (18.5 to $<25 \, kg/m^2$), overweight (25 to $<30 \, kg/m^2$) and obese ($\geq 30 \, kg/m^2$). We excluded participants with a BMI in the underweight range because there were insufficient

numbers for meaningful analysis as a separate group (n=64, 1.2% of otherwise eligible sample).

For some descriptive analyses, normal-weight participants were divided into those with a BMI in the lower half of the normal-weight range (BMI 18.5 to $<21.75 \, kg/m^2$; 'lower normal-weight') and those in the upper half (BMI 21.75 to $<30 \, kg/m^2$; 'upper normal-weight') to provide an indication of the distribution of participants across the normal-weight range and help interpret associations between BMI and weight perceptions.

### Sociodemographic information
Information on age, sex, ethnicity (white vs non-white) and socioeconomic status (SES) was recorded. SES was indexed using the short version of the National Statistics Socio-Economic Classification 3 category classification of the present or most recent occupation and categorised as higher (managerial/professional occupations), intermediate (intermediate occupations) and lower (routine/manual occupations).[19] This measure of SES was chosen for comparability with previous studies investigating weight perceptions in other age groups in England.[20]

### Perceived health and comorbidities
Self-rated health was assessed using a single item: '*Would you say your health is… poor/fair/good/very good/excellent?*' We analysed the proportion of individuals rating their health as fair/poor. This dichotomy is commonly used in analyses of this variable[21–23] to overcome issues relating to the skewed distribution of responses and provide results that are easily interpreted (ie, odds of the outcome associated with poorer vs better health).

Information about five doctor-diagnosed chronic diseases that may cause weight loss (cancer, stroke, chronic lung disease, diabetes and arthritis) was self-reported and the number of reported conditions was summed to create a chronic health condition index ranging from 0 to 5. Because scores were highly skewed, we dichotomised this variable to distinguish between 0 and ≥1 health conditions.

### Statistical analysis
Analyses were performed using IBM SPSS Statistics V.24. In order to produce representative estimates for older adults in the English population, data were weighted to correct for sampling probabilities and to match the English population on age and sex. The weights accounted for the differential probability of being included in Wave 8 of ELSA.

Analyses were performed separately for men and women. We tested sex differences in sociodemographic, anthropometric, self-rated health, and weight perception variables using t-tests for continuous variables and Pearson's $\chi^2$ analyses for categorical variables. We used multivariable logistic regression to identify independent associations with perception of weight as (1) 'too heavy' or (2) 'too light' among normal-weight participants, and associations with perception of weight as 'about right'

**Table 1** Sample demographic and anthropometric characteristics

| | Whole sample (n=5240) | Men (n=2352) | Women (n=2888) | P value* |
|---|---|---|---|---|
| Age (mean (SD) years) | 70.07 (7.76) | 70.23 (7.52) | 70.57 (7.71) | 0.117 |
| Ethnicity | | | | |
| White | 96.2 | 95.8 | 96.6 | 0.163 |
| Non-white | 3.8 | 4.2 | 3.4 | – |
| SES | | | | |
| Higher | 32.4 | 38.8 | 26.6 | <0.001 |
| Intermediate | 26.8 | 21.9 | 31.3 | – |
| Lower | 40.8 | 39.4 | 42.2 | – |
| Self-rated health | | | | |
| Good/very good/excellent | 72.7 | 72.5 | 72.9 | 0.790 |
| Fair/poor | 27.3 | 27.5 | 27.1 | – |
| Chronic health conditions | | | | |
| None | 42.0 | 46.2 | 38.1 | <0.001 |
| ≥1 | 58.0 | 53.8 | 61.9 | – |
| Height (mean (SD) cm) | 166.31 (9.44) | 173.10 (6.90) | 160.01 (6.66) | <0.001 |
| Weight (mean (SD) kg) | 78.64 (16.35) | 84.59 (15.04) | 72.81 (15.29) | <0.001 |
| BMI (mean (SD) kg/m$^2$) | 28.39 (5.28) | 28.21 (4.61) | 28.45 (5.76) | 0.116 |
| Weight status | | | | |
| Normal weight | 27.5 | 23.8 | 30.8 | <0.001 |
| Lower normal weight | 7.1 | 4.7 | 9.2 | – |
| Upper normal weight | 20.4 | 19.1 | 21.5 | – |
| Overweight | 40.3 | 46.1 | 35.0 | – |
| Obese | 32.3 | 30.1 | 34.2 | – |

Weighted means and proportions are shown. Sample sizes (n) are shown unweighted.

*P values are for the association between each variable and sex.

SES, socioeconomic status; BMI, body mass index.

among participants with overweight or obesity. Variables included in the models were age, ethnicity, SES, self-rated health and BMI. In order to evaluate whether perception of weight was too light among normal-weight participants, or about right among participants with overweight or obesity, was associated with health conditions that may cause weight loss, we also adjusted for chronic health conditions in each model. Results are reported as ORs with 95% CIs.

### Patient and public involvement

Patients and the public were not involved in this research.

### RESULTS

There were 2352 men and 2888 women in the sample. Descriptive characteristics overall and by sex are shown in table 1. The mean age was 70.1 (SD 7.8) years. The majority of participants were white (96.2%) and rated their health as good, very good or excellent (72.7%) despite many having one or more chronic health conditions (58.0%). There was fairly even distribution across socioeconomic groups (32.4% higher SES, 26.8% intermediate SES, 40.8% lower SES), although women were less likely than men to be in the higher SES group (26.6% vs 38.8%, p<0.001). Just over a quarter (27.5%) of participants had a BMI in the normal-weight range, 40.3% had overweight and a further 32.3% had obesity. On average, men were significantly taller and heavier than women. While there was no significant sex difference in mean BMI, men were more likely than women to have a BMI placing them in the overweight range (46.1% vs 35.0%) while women were more likely than men to have a normal-weight (30.8% vs 23.8%) or obese BMI (34.2% vs 30.1%).

### Weight perception among normal-weight older adults

Figure 1 summarises the distribution of weight perceptions by sex and weight status. In the normal-weight category, the majority (80.3%) of participants thought they were about the right weight, but 9.9% thought they were too light, and 9.7% thought they were too heavy. Normal-weight women were significantly more likely than normal-weight men to consider themselves to be too heavy (12.6%

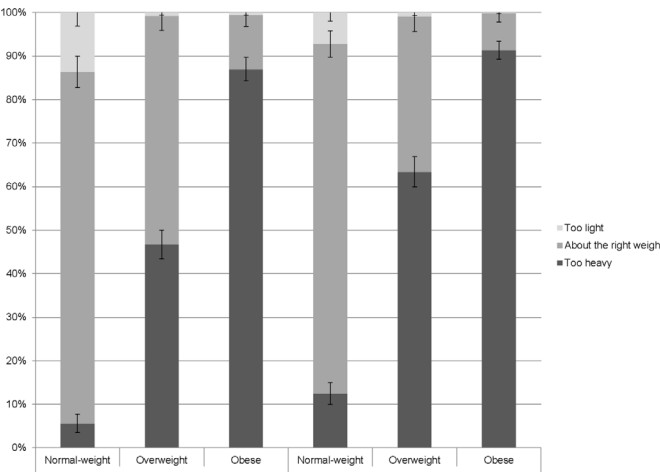

**Figure 1** The proportion (with 95% CI) of men and women who reported feeling 'too heavy', 'about the right weight' and 'too light', by measured weight status. Weighted data shown.

vs 5.6%, p<0.001) and less likely to consider themselves too light (7.3% vs 13.7%, p<0.001).

Multivariable models testing independent associations of age, sex, ethnicity, SES and BMI with perception of body weight as too heavy in normal-weight men and women are presented in table 2 (left panel). There was a significant independent association between BMI and perception of weight as too heavy in both sexes, with each unit increase in BMI associated with 1.76 times higher odds (95% CI 1.18 to 2.61) of perception of weight as too heavy in men and 2.14 times higher odds (95% CI 1.69 to 2.72) in women. Some 6.7% of men and 16.8% of women in the upper normal-weight group felt too heavy, compared with just 1.1% of men and 2.5% of women in the lower normal-weight group. Older age was significantly associated with reduced odds of perception of weight as too heavy in women (OR=0.97, 95% CI 0.94 to 1.00) but not in men. There was no significant independent association between perception of weight as too heavy and ethnicity, SES or self-rated health in either sex.

Factors independently associated with perception of body weight as too light in normal-weight men and women are shown in table 2 (right panel). There were strong independent associations with BMI and self-rated health. In both sexes, odds of feeling too light were significantly lower among those with a higher BMI, with each unit increase in BMI associated with a 47% reduction in odds in men (OR=0.53, 95% CI 0.43 to 0.64) and a 53% reduction in odds in women (OR=0.47, 95% CI 0.38 to 0.58). Just 6.5% of men and 2.6% of women in the upper normal-weight group felt too light, compared with 43.3% of men and 17.8% of women in the lower normal-weight group. Odds of perception of weight as too light were significantly increased among participants with fair/poor self-rated health (men: OR=3.70, 95% CI 1.88 to 7.28; women: OR=2.61, 95% CI 1.27 to 5.35). The presence of one or more chronic health conditions was also independently associated with increased odds of

perception of weight as too light in women (OR=2.21, 95% CI 1.03 to 4.77) but not in men. Older age was associated with increased odds of perception of weight as too light in normal-weight women (OR=1.04, 95% CI 1.01 to 1.09) but not in men. Non-white ethnicity was associated with increased odds of perception of weight as too light in normal-weight men (OR=6.26, 95% CI 2.09 to 18.76) but not in women. There was no significant independent association with SES in either sex.

## Weight perception among older adults with overweight or obesity

Among participants with a BMI in the overweight range, 54.3% thought they were too heavy, but 44.8% thought they were about the right weight and 0.9% thought they were too light. Among participants with an obese BMI, 89.4% thought they were too heavy, 10.3% thought they were about the right weight and 0.4% thought they were too light. Women with overweight were more likely than men with overweight to recognise that they were too heavy (63.4% vs 46.7%, p<0.001) and less likely to perceive themselves to be about the right weight (35.7% vs 52.5%, p<0.001). Likewise, women with obesity were more likely than men with obesity to perceive their weight as too heavy (91.3% vs 87.0%, p=0.012) and less likely to perceive their weight as about right (8.5% vs 12.4%, p=0.020) (figure 1).

Factors independently associated with perception of body weight as about right in men and women with overweight are summarised in table 3. Among men and women with overweight, there was a strong independent association with BMI: each unit increase in BMI was associated with 48% lower odds of perception of weight as about right in men (OR=0.52, 95% CI 0.47 to 0.59) and 45% lower odds in women (OR=0.55, 95% CI 0.48 to 0.63). Older age was also independently associated with increased odds of perception of weight as about right in both sexes (men: OR=1.04, 95% CI 1.02 to 1.06; women: OR=1.06, 95% CI 1.04 to 1.09). Non-white ethnicity was associated with increased odds of perception of weight as about right in women (OR=3.79, 95% CI 1.44 to 9.94) but not in men. An association with SES was also observed in women, with women with overweight in the intermediate and lower SES groups significantly more likely to perceive their weight to be about right than those from the higher SES group (intermediate: OR=1.78, 95% CI 1.14 to 2.79; lower: OR=1.73, 95% CI 1.13 to 2.66). There was no significant association with SES in men. There was no significant independent association with self-rated health in either sex, although in men (but not in women), presence of at least one comorbid health condition was associated with significantly lower odds of perception of weight as about right (OR=0.56, 95% CI 0.41 to 0.76).

Factors independently associated with perception of body weight as about right in men and women with obesity are shown in table 4. Older age was significantly associated with increased odds of perception of body weight as about right in both men (OR=1.05, 95% CI 1.01 to 1.09)

**Table 2** Multivariable models testing associations with feeling 'too heavy' and 'too light' among normal-weight older men and women

| | Too heavy | | | | | | Too light | | | | | |
| | Men (n=581) | | | Women (n=917) | | | Men (n=581) | | | Women (n=917) | | |
| | %* | OR (95% CI) | P value | %* | OR (95% CI) | P value | %* | OR (95% CI) | P value | %* | OR (95% CI) | P value |
|---|---|---|---|---|---|---|---|---|---|---|---|---|
| Age (years) | – | 1.01 (0.96 to 1.06) | 0.717 | – | 0.97 (0.94 to 1.00) | 0.049 | – | 1.01 (0.97 to 1.05) | 0.745 | – | 1.04 (1.00 to 1.09) | 0.042 |
| Ethnicity | | | | | | | | | | | | |
| White | 5.7 | 1.00 | – | 12.6 | 1.00 | – | 12.2 | 1.00 | – | 7.1 | 1.00 | – |
| Non-white | 4.8 | 0.94 (0.14 to 6.55) | 0.952 | 9.5 | 0.42 (0.09 to 1.89) | 0.258 | 42.9 | 6.26 (2.09 to 18.76) | 0.001 | 9.5 | 1.96 (0.36 to 10.82) | 0.440 |
| SES | | | | | | | | | | | | |
| Higher | 5.4 | 1.00 | – | 11.9 | 1.00 | – | 10.3 | 1.00 | – | 3.7 | 1.00 | – |
| Intermediate | 7.8 | 1.40 (0.53 to 3.71) | 0.504 | 12.5 | 1.18 (0.64 to 2.17) | 0.605 | 10.7 | 0.84 (0.36 to 1.96) | 0.689 | 9.3 | 2.26 (0.87 to 5.86) | 0.093 |
| Lower | 5.1 | 1.03 (0.40 to 2.70) | 0.948 | 13.4 | 1.26 (0.69 to 2.31) | 0.457 | 19.7 | 1.30 (0.64 to 2.64) | 0.474 | 8.9 | 2.15 (0.83 to 5.57) | 0.114 |
| Self-rated health | | | | | | | | | | | | |
| Good/very good/excellent | 5.6 | 1.00 | – | 13.1 | 1.00 | – | 8.2 | 1.00 | – | 4.8 | 1.00 | – |
| Fair/poor | 5.8 | 1.17 (0.47 to 2.92) | 0.730 | 9.9 | 0.88 (0.45 to 1.72) | 0.717 | 29.2 | 3.70 (1.88 to 7.28) | <0.001 | 17.6 | 2.61 (1.27 to 5.35) | 0.009 |
| BMI | – | 1.76 (1.18 to 2.61) | 0.005 | – | 2.14 (1.69 to 2.72) | <0.001 | – | 0.53 (0.43 to 0.64) | <0.001 | – | 0.47 (0.38 to 0.58) | <0.001 |
| Chronic health conditions | | | | | | | | | | | | |
| None | – | – | – | – | – | – | 10.0 | 1.00 | – | 4.3 | 1.00 | – |
| ≥1 | – | – | – | – | – | – | 17.3 | 1.07 (0.54 to 2.14) | 0.840 | 10.0 | 2.21 (1.03 to 4.77) | 0.043 |

Weighted data. Sample sizes (n) are shown unweighted.

*Indicates the percentage of normal-weight participants in each group perceiving themselves to be too heavy/too light.

SES, socioeconomic status; BMI, body mass index.

**Table 3** Multivariable models testing associations with feeling 'about the right weight' among older men and women with overweight

| | Men (n=1083) | | | Women (n=1052) | | |
|---|---|---|---|---|---|---|
| | %* | OR (95% CI) | P value | %* | OR (95% CI) | P value |
| Age (years) | – | 1.04 (1.02 to 1.06) | <0.001 | – | 1.06 (1.04 to 1.09) | <0.001 |
| Ethnicity | | | | | | |
| White | 52.7 | 1.00 | – | 34.9 | 1.00 | – |
| Non-white | 48.8 | 1.77 (0.87 to 3.59) | 0.115 | 61.9 | 3.79 (1.44 to 9.94) | 0.007 |
| SES | | | | | | |
| Higher | 53.5 | 1.00 | – | 28.9 | 1.00 | – |
| Intermediate | 54.4 | 1.22 (0.83 to 1.81) | 0.313 | 37.5 | 1.78 (1.14 to 2.79) | 0.011 |
| Lower | 50.4 | 1.14 (0.81 to 1.59) | 0.458 | 38.1 | 1.73 (1.13 to 2.66) | 0.013 |
| Self-rated health | | | | | | |
| Good/very good/excellent | 54.3 | 1.00 | – | 34.1 | 1.00 | – |
| Fair/poor | 46.4 | 0.93 (0.65 to 1.33) | 0.688 | 40.3 | 1.26 (0.84 to 1.89) | 0.273 |
| BMI | – | 0.52 (0.47 to 0.59) | <0.001 | – | 0.55 (0.48 to 0.63) | <0.001 |
| Chronic health conditions | | | | | | |
| None | 58.8 | 1.00 | – | 32.6 | 1.00 | – |
| ≥1 | 45.8 | 0.56 (0.41 to 0.76) | <0.001 | 37.4 | 1.08 (0.76 to 1.55) | 0.659 |

Weighted data. Sample sizes (n) are shown unweighted.
*Indicates the percentage of overweight participants in each group perceiving themselves to be about the right weight or too light.
BMI, body mass index; SES, socioeconomic status.

and women (OR=1.04, 95% CI 1.01 to 1.08). Higher BMI was associated with 24% lower odds of perception of body weight as about right in men (OR=0.76, 95% CI 0.67 to 0.87) but no significant association with BMI was observed in women. Non-white ethnicity was significantly associated with increased odds of perception of body

**Table 4** Multivariable models testing associations with feeling 'about the right weight' among older men and women with obesity

| | Men (n=688) | | | Women (n=919) | | |
|---|---|---|---|---|---|---|
| | %* | OR (95% CI) | P value | %* | OR (95% CI) | P value |
| Age (years) | – | 1.05 (1.01 to 1.09) | 0.013 | – | 1.04 (1.01 to 1.08) | 0.012 |
| Ethnicity | | | | | | |
| White | 12.7 | 1.00 | – | 8.1 | 1.00 | – |
| Non-white | 5.3 | 0.32 (0.04 to 2.48) | 0.278 | 20.7 | 2.76 (1.03 to 7.40) | 0.043 |
| SES | | | | | | |
| Higher | 11.1 | 1.00 | – | 8.0 | 1.00 | – |
| Intermediate | 11.8 | 0.96 (0.46 to 1.98) | 0.901 | 4.5 | 0.56 (0.23 to 1.34) | 0.190 |
| Lower | 13.8 | 1.34 (0.74 to 2.43) | 0.343 | 11.0 | 1.46 (0.74 to 2.87) | 0.272 |
| Self-rated health | | | | | | |
| Good/very good/excellent | 13.1 | 1.00 | – | 8.3 | 1.00 | – |
| Fair/poor | 11.3 | 0.98 (0.55 to 1.75) | 0.935 | 9.2 | 1.01 (0.56 to 1.83) | 0.968 |
| BMI | – | 0.76 (0.67 to 0.87) | <0.001 | – | 0.96 (0.90 to 1.03) | 0.269 |
| Chronic health conditions | | | | | | |
| None | 13.9 | 1.00 | – | 8.6 | 1.00 | – |
| ≥1 | 11.6 | 0.84 (0.48 to 1.47) | 0.541 | 8.5 | 0.81 (0.43 to 1.52) | 0.509 |

Weighted data. Sample sizes (n) are shown unweighted.
*Indicates the percentage of overweight participants in each group perceiving themselves to be about the right weight or too light.
BMI, body mass index; SES, socioeconomic status.

weight as about right in women (OR=2.76, 95% CI 1.03 to 7.40) but not in men. No significant association between perception of body weight as about right and SES, self-rated health or chronic health conditions was observed in either sex.

## DISCUSSION

In a large, representative sample of older adults living in England, we found that weight perceptions broadly corresponded to participants' actual weight status as defined by widely used BMI cut-offs: 80% of older adults with a BMI in the normal-weight range thought they were about the right weight, and over 50% of older adults with a BMI in the overweight range and almost 90% of those with a BMI in the obese range thought they were too heavy. However, a substantial number of older adults either underestimated or overestimated their weight status relative to their BMI category. One in 10 participants with a normal-weight BMI thought they were too heavy and 1 in 10 thought they were too light. Almost half of participants with an overweight BMI thought they were about the right weight, as did 1 in 10 of those with an obese BMI.

As has been observed in previous research examining weight perceptions in younger samples,[1 20 24] there were systematic differences between men and women's weight perceptions. Across all BMI categories, men were consistently more likely than women to underestimate their weight status, and normal-weight women were more likely than normal-weight men to report feeling too heavy. Within BMI categories, those with a higher BMI were more likely to perceive themselves to be too heavy and less likely to perceive themselves to be too light or about right. There were also some differences in weight perceptions by ethnicity and SES, with people from non-white ethnic groups and intermediate and lower SES groups more likely to underestimate their weight status, as has been shown previously in younger samples,[20 24] although these differences were not consistently observed across BMI categories or sexes.

Importantly, there were also clear age-related differences in weight perceptions across all BMI categories. In men and women with an overweight or obese BMI, the odds of feeling about the right weight increased with advancing age. In women with a normal-weight BMI, the odds of feeling too light increased and the odds of feeling too heavy decreased with age, although there were no significant differences by age in men. These findings are suggestive of a higher ideal weight at older ages. This is consistent with previous studies that have shown people, particularly women, tend to endorse a slightly larger and more curvaceous body shape as they get older.[8 25] It is possible that older people believe that having an overweight or obese BMI is not necessarily a bad thing at older ages.[12] Alternatively, as people get older they may give up the effort to reduce weight at higher BMIs, perhaps because other health issues become more salient for them, or because they are less interested in self-presentation and striving for a slimmer physique.[12]

There was also a significant association between self-rated health and odds of feeling too light among men and women with a normal-weight BMI. Those who rated their health as fair or poor had around three times higher odds of feeling too light at a normal-weight BMI than those who rated their health as good, very good or excellent. In addition, women with a normal-weight BMI who had at least one chronic condition were also more likely to consider themselves to be too light, although there was no difference in men. These results could be interpreted as suggesting that the ideal weight may be higher for older adults in poorer health than for those in good health. There is a vast literature documenting an obesity paradox in chronic conditions including cardiovascular disease,[26] cancer,[27] kidney disease[28] and lung disease,[29] whereby patients who carry excess weight have better outcomes than those with a normal-weight BMI. It has also been suggested that stress related to negative body image perception may have a causal role in the development of poor health.[30] An alternative explanation is that the association between poorer self-rated health and increased likelihood of feeling too light might be secondary to illness. Older people with serious chronic illnesses often lose weight, so the fact they think they are too light might be a reflection of this concern.

At the upper end of the weight spectrum, health status was less strongly associated with weight perceptions, with no significant association observed between self-rated health and perception of weight as about right among those with an overweight or obese BMI. However, men with a BMI in the overweight range who had at least one comorbid condition had significantly lower odds of feeling their weight was about right than those who were free of comorbidities, suggesting that experiencing an associated health problem may make men more likely to recognise their overweight (although still less likely than women with or without a health condition).

Taken together, these findings suggest that the discrepancy between perceived weight status and BMI-based definitions of weight status is greater among people who are older or in poorer health. The appropriateness of conventional BMI definitions of weight status for older people is an issue of ongoing debate in the literature.[31–35] Currently, guidance issued by official bodies such as the UK National Health Service[36] and US Centers for Disease Control and Prevention[37] does not differ according to age group or health status. However, many health professionals are reluctant to recommend weight loss for older patients with an overweight BMI[38 39] and there have been calls to reconsider the standards for ideal weight at older ages and develop age-specific recommendations.[40–42] The present results show that for the majority of older people, weight perceptions map onto BMI definitions of weight status. Further research is needed to establish whether for the remainder, the mismatch between perceived weight and BMI status represents a lack of awareness of 'healthy'

weight, preference for higher weight in older age, or reflects a genuine biological advantage to being heavier in older age (the so-called 'obesity paradox').[35]

The strengths of the present study include the large sample size and objective measurements of height and weight. However, findings should be considered in the light of several limitations. There was a substantial amount of missing data, so findings may not generalise to the entire population. Of note, weight measurements were not available for all Wave 8 participants (8% missing). If those who were more concerned about their weight were more likely to decline to be measured, our results may underestimate the proportion of older adults across all weight groups who consider themselves to be too heavy. In addition, no data were collected on participants' weight history, which could potentially influence their current perceptions of their body weight (eg, individuals with a higher average lifetime BMI or history of overweight or obesity may be more likely to incorrectly perceive themselves to be 'too light' or 'about right' in older age).

In conclusion, the present results indicate that older adults' perceptions of their own weight generally correspond with traditional BMI cut-offs for normal weight, overweight and obesity. However, a substantial minority 'underestimate' their weight status, with those at the upper end of the age spectrum and those in poorer health more likely to do so.

**Contributors** SEJ analysed the data and drafted the manuscript. LS and AS provided critical revisions and approved the final manuscript. All researchers listed as authors are independent from the funders and all final decisions about the research were taken by the investigators and were unrestricted. All authors had full access to all of the data (including statistical reports and tables) in the study and can take responsibility for the integrity of the data and the accuracy of the data analysis.

**Funding** This work was supported by Cancer Research UK (C1417/A22962) and the ESRC (ES/R005990/1).

**Disclaimer** The funders had no final role in the study design; in the collection, analysis and interpretation of data; in the writing of the report; or in the decision to submit the paper for publication.

**Competing interests** None declared.

**Patient consent for publication** Not required.

**Provenance and peer review** Not commissioned; externally peer reviewed.

**Data availability statement** Data are available in a public, open access repository. The raw ELSA data are available from the UK Data Service.

**ORCID iD**
Sarah E Jackson http://orcid.org/0000-0001-5658-6168

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
