## [Reviewer comments · BMJ Open]

ARTICLE DETAILS

TITLE (PROVISIONAL)	Weight perceptions in older adults: findings from the English Longitudinal Study of Ageing
AUTHORS	Jackson, Sarah; Smith, Lee; Steptoe, Andrew

VERSION 1 - REVIEW

REVIEWER	Ashleigh Haynes Cancer Council Victoria, Australia
REVIEW RETURNED	10-Oct-2019

GENERAL COMMENTS	Thank you for the opportunity to review this manuscript. The manuscript is clearly written and reports an analysis of factors associated with weight perception in a representative sample of older adults in England, in particular, self-rated health. The introduction touches on various physiological and social age-related changes that may underlie the shift in weight perceptions in older age. The results are interpreted as consistent with other literature questioning the use of conventional BMI definitions of weight status in older adults. The authors review evidence to suggest that higher BMI is paradoxically associated with better health outcomes in older adults (which in itself gives reason to reconsider the conventional BMI cutoffs), but a clearer rationale for why under perception among those with poorer self-rated health supports the need to reconsider BMI criteria is needed. I wasn't entirely convinced by the treatment of this argument in the introduction and discussion as it stands. I have only a few further suggestions: - page 2, line 18: "...low sensitivity in identifying older adults at risk of undernutrition or obesity " would be helpful to specify here the criterion used to diagnose obesity in these studies (in comparison to body mass index)- Re: the choice of SES measure. Was this the only SES measure taken in this wave of ELSA? If not, why was this current/former occupation measure chosen over other potential indicators or a combination of them (e.g., income, education, postcode?)- The analyses adjust for chronic health conditions, but only in the 'normal weight' models, with the reasoning that these conditions can cause weight loss. Why not adjust for chronic health conditions in overweight and obesity models? That chronic health conditions may be associated with weight loss and therefore promote under perception also applies to individuals with overweight and obesity (despite weight loss, these individuals may
---

	still fall within the currently defined 'overweight' or 'obesity' classes).  - Along these lines, does the ELSA have data on participants' former weight or highest lifetime weight etc? If so, the authors might consider including it in models. At older age, weight perception may be influenced by age-related bodily changes as well one's weight history. Regardless, commenting on former or lifetime weight in the discussion would be useful, as this could potentially affect weight perception (e.g., individuals with a higher average lifetime BMI, or history of overweight or obesity may be more likely to incorrectly perceive themselves as 'too light' or 'about right' in older age). - Perceived health: the authors report that they dichotomise this variable because this is the way it is commonly treated – why is it commonly treated in this way and why is this more suitable than other potential approaches? - The authors use first-person language throughout most but not all of the manuscript (e.g., line 6, page 7; line 46 page 8). - The sample sizes reported at line 24, page 7 are different from those reported in Table 1. This may well be a weighting issue, but the authors should either clarify in the manuscript the reason for, or correct, the discrepancy. - A potential rounding error in the reporting of mean age? - The authors report descriptive statistics for participants in the lower and upper-normal weight ranges separately, but do not later comment on these results. What is the purpose of their inclusion? Briefly touching on this in the discussion would be useful.
--	--

REVIEWER	Dr. Michael Daly Maynooth University, Ireland
REVIEW RETURNED	22-Oct-2019

GENERAL COMMENTS	This interesting study draws on a large representative sample of older English adults to understand patterns of weight perception and their relation to objective weight status and demographic factors and health ratings and conditions. Understanding the nature of weight status perceptions among older adults, given their relation to a broad set of health-related outcomes (e.g. Haynes et al., 2018), is certainly a worthwhile area of research. However, in my view the focus of the article on reconsidering BMI moves too far beyond what it is possible to say using the data, particularly given that weight perceptions are determined by many social and cultural factors as acknowledged in the article. If I may, I would suggest refocusing the paper on understanding weight perception in older adults rather than on the limitations of BMI (which is perhaps more of a discussion point given the study findings).  - The research question and study objectives are unclear. The title poses the provocative question “Time to reconsider body mass index for defining weight status in old age?” yet the study aims to “explore perceptions of weight status” as compared with objective weight status as has been done in a range of studies including in the UK (Robinson & Oldham, 2016). How does considering weight perceptions lead us to reconsider BMI as our key method for defining weight status?
--

	- There is evidence that the obesity paradox may reflect methodological problems in epidemiological studies including aspects of selection bias such as collider bias (e.g. Lennon, Sperrin, Badrick & Renehan, 2016; Vansteelandt, 2017). As such, it is not clear (at least from the introduction section) that the literature has reached the point where we can say that the obesity paradox is a genuine phenomenon in that obesity, as currently, defined, may have a causal protective effect or require redefinition for this reason. It is also unclear from the introduction what insight specifically we would gain into the obesity paradox by examining weight perception. - The logic underlying the key statement below requires some expansion given the possibility for third variables to lead to this relation (e.g. low health literacy leads to both poor health and body weight underestimation). “If underestimation of body weight is more pronounced for those who perceive their health to be poor, it would suggest that the optimal BMI for older people may be higher than for younger adults and further call into question the suitability of existing BMI cut-offs to define weight status in later life.” Haynes, A., Kersbergen, I., Sutin, A., Daly, M., & Robinson, E. (2018). A systematic review of the relationship between weight status perceptions and weight loss attempts, strategies, behaviours and outcomes. Obesity reviews, 19(3), 347-363. Lennon, H., Sperrin, M., Badrick, E., & Renehan, A. G. (2016). The obesity paradox in cancer: a review. Current oncology reports, 18(9), 56. Robinson, E., & Oldham, M. (2016). Weight status misperceptions among UK adults: the use of self-reported vs. measured BMI. BMC obesity, 3(1), 21. Vansteelandt, S. (2017). Asking too much of epidemiologic studies: the problem of collider bias and the obesity paradox. Epidemiology, 28(5), e47-e49.
--	--

VERSION 1 – AUTHOR RESPONSE

Reviewer: 1

Thank you for the opportunity to review this manuscript. The manuscript is clearly written and reports an analysis of factors associated with weight perception in a representative sample of older adults in England, in particular, self-rated health. The introduction touches on various physiological and social age-related changes that may underlie the shift in weight perceptions in older age. The results are interpreted as consistent with other literature questioning the use of conventional BMI definitions of weight status in older adults. The authors review evidence to suggest that higher BMI is paradoxically associated with better health outcomes in older adults (which in itself gives reason to reconsider the conventional BMI cutoffs), but a clearer rationale for why under perception among those with poorer self-rated health supports the need to reconsider BMI criteria is needed. I wasn't entirely convinced by the treatment of this argument in the introduction and discussion as it stands.

Response: We appreciate these comments. Given the views of both reviewers, we have chosen to reframe the paper taking focus away from the obesity paradox and concentrating on understanding older adults' weight perceptions in and of themselves.

I have only a few further suggestions:

- page 2, line 18: "...low sensitivity in identifying older adults at risk of undernutrition or obesity " would be helpful to specify here the criterion used to diagnose obesity in these studies (in comparison to body mass index)

Response: We have clarified that the cited studies suggesting BMI showed low sensitivity in identifying obesity used dual energy X-ray absorptiometry measurement of body fat percentage.

- Re: the choice of SES measure. Was this the only SES measure taken in this wave of ELSA? If not, why was this current/former occupation measure chosen over other potential indicators or a combination of them (e.g., income, education, postcode?)

Response: We have added a sentence explaining our rationale:

"This measure of SES was chosen for comparability with previous studies investigating weight perceptions in other age groups in England e.g. 28."

- The analyses adjust for chronic health conditions, but only in the 'normal weight' models, with the reasoning that these conditions can cause weight loss. Why not adjust for chronic health conditions in overweight and obesity models? That chronic health conditions may be associated with weight loss and therefore promote under perception also applies to individuals with overweight and obesity (despite weight loss, these individuals may still fall within the currently defined 'overweight' or 'obesity' classes).

Response: This is a good point. We have added chronic health conditions into the models for overweight and obesity and updated results in the text and tables 3 and 4. There was a significant association between presence of chronic health conditions and lower odds of perception of weight as about right in men with BMIs in the overweight range. Chronic health conditions were not associated with weight perceptions in women with overweight or men or women with obesity. We've added the following to the discussion:

"At the upper end of the weight spectrum, health status was less strongly associated with weight perceptions, with no significant association observed between self-rated health and perception of weight as about right among those with an overweight or obese BMI. However, men with a BMI in the overweight range who had at least one comorbid condition had significantly lower odds of feeling their weight was about right than those who were free of comorbidities, suggesting that experiencing an associated health problem may make men more likely to recognise their overweight (albeit still less likely than women with or without a health condition)."

- Along these lines, does the ELSA have data on participants' former weight or highest lifetime weight etc? If so, the authors might consider including it in models. At older age, weight perception may be influenced by age-related bodily changes as well one's weight history. Regardless, commenting on former or lifetime weight in the discussion would be useful, as this could potentially affect weight perception (e.g., individuals with a higher average lifetime BMI, or history of overweight or obesity may be more likely to incorrectly perceive themselves as 'too light' or 'about right' in older age).

Response: Unfortunately, the ELSA survey does not include information on weight history. We now raise this as a limitation in the discussion:

“In addition, no data were collected on participants’ weight history, which could potentially influence their current perceptions of their body weight (for example, individuals with a higher average lifetime BMI or history of overweight or obesity may be more likely to incorrectly perceive themselves to be ‘too light’ or ‘about right’ in older age).”

- Perceived health: the authors report that they dichotomise this variable because this is the way it is commonly treated – why is it commonly treated in this way and why is this more suitable than other potential approaches?

Response: We have added the following to explain:

“This dichotomy is commonly used in analyses of this variable 29–31 to overcome issues relating to the skewed distribution of responses and provide results that are easily interpreted (i.e. odds of the outcome associated with poorer versus better health).”

- The authors use first-person language throughout most but not all of the manuscript (e.g., line 6, page 7; line 46 page 8).

Response: Thank you for picking this up. We have corrected to people-first language.

- The sample sizes reported at line 24, page 7 are different from those reported in Table 1. This may well be a weighting issue, but the authors should either clarify in the manuscript the reason for, or correct, the discrepancy.

Response: Thank you for picking up this error. The sample sizes reported in the text were incorrect, we have corrected to match the table and total sample size reported in the abstract and method.

- A potential rounding error in the reporting of mean age?

Response: We’ve corrected this.

- The authors report descriptive statistics for participants in the lower and upper-normal weight ranges separately, but do not later comment on these results. What is the purpose of their inclusion? Briefly touching on this in the discussion would be useful.

Response: We now explain in the method why we include these descriptive data:

“For some descriptive analyses, normal-weight participants were divided into those with a BMI in the lower half of the normal-weight range (BMI 18.5 to <21.75 kg/m²; ‘lower normal-weight’) and those in the upper half (BMI 21.75 to <30 kg/m²; ‘upper normal-weight’) to provide an indication as to the distribution of participants across the normal-weight range and help interpret associations between BMI and weight perceptions.”

We touch on within BMI category differences in weight perceptions in the discussion:

“Within BMI categories, those with a higher BMI were more likely to perceive themselves to be too heavy and less likely to perceive themselves to be too light or about right.”

Reviewer: 2

This interesting study draws on a large representative sample of older English adults to understand patterns of weight perception and their relation to objective weight status and demographic factors and health ratings and conditions. Understanding the nature of weight status perceptions among older adults, given their relation to a broad set of health-related outcomes (e.g. Haynes et al., 2018), is certainly a worthwhile area of research. However, in my view the focus of the article on reconsidering BMI moves too far beyond what it is possible to say using the data, particularly given that weight perceptions are determined by many social and cultural factors as acknowledged in the article. If I may, I would suggest refocusing the paper on understanding weight perception in older adults rather than on the limitations of BMI (which is perhaps more of a discussion point given the study findings).

Response: We appreciate your comments and have taken on board your suggestion to refocus the paper. We now frame it with a focus on understanding older adults' weight perceptions, mentioning the limitations of BMI in the discussion, as suggested:

"Taken together, these findings suggest that the discrepancy between perceived weight status and BMI-based definitions of weight status is greater among people who are older or in poorer health. The appropriateness of conventional BMI definitions of weight status for older people is an issue of ongoing debate in the literature 31–35. Currently, guidance issued by official bodies such as the UK National Health Service 36 and US Centers for Disease Control and Prevention 37 does not differ according to age group or health status. However, many health professionals are reluctant to recommend weight loss for older patients with an overweight BMI 38,39 and there have been calls to reconsider the standards for ideal weight at older ages and develop age-specific recommendations 40–42. The present results show that for the majority of older people, weight perceptions map onto BMI definitions of weight status. Further research is needed to establish whether for the remainder, the mismatch between perceived weight and BMI status represents a lack of awareness of 'healthy' weight, preference for higher weight in older age, or reflects a genuine biological advantage to being heavier in older age (the so-called 'obesity paradox' 35)."

- The research question and study objectives are unclear. The title poses the provocative question "Time to reconsider body mass index for defining weight status in old age?" yet the study aims to "explore perceptions of weight status" as compared with objective weight status as has been done in a range of studies including in the UK (Robinson & Oldham, 2016). How does considering weight perceptions lead us to reconsider BMI as our key method for defining weight status?

Response: We have now changed the title to more accurately reflect the aims of these analyses: "Weight perceptions in older adults: findings from the English Longitudinal Study of Ageing"

- There is evidence that the obesity paradox may reflect methodological problems in epidemiological studies including aspects of selection bias such as collider bias (e.g. Lennon, Sperrin, Badrick & Renehan, 2016; Vansteelandt, 2017). As such, it is not clear (at least from the introduction section) that the literature has reached the point where we can say that the obesity paradox is a genuine phenomenon in that obesity, as currently, defined, may have a causal protective effect or require redefinition for this reason. It is also unclear from the introduction what insight specifically we would gain into the obesity paradox by examining weight perception.

Response: In line with your suggestion, we have reframed the introduction removing the focus on the obesity paradox and discuss the results in relation to BMI much more cautiously:

“The present results show that for the majority of older people, weight perceptions map onto BMI definitions of weight status. Further research is needed to establish whether for the remainder, the mismatch between perceived weight and BMI status represents a lack of awareness of ‘healthy’ weight, preference for higher weight in older age, or reflects a genuine biological advantage to being heavier in older age (the so-called ‘obesity paradox’ 35).”

- The logic underlying the key statement below requires some expansion given the possibility for third variables to lead to this relation (e.g. low health literacy leads to both poor health and body weight underestimation).

“If underestimation of body weight is more pronounced for those who perceive their health to be poor, it would suggest that the optimal BMI for older people may be higher than for younger adults and further call into question the suitability of existing BMI cut-offs to define weight status in later life.”

Haynes, A., Kersbergen, I., Sutin, A., Daly, M., & Robinson, E. (2018). A systematic review of the relationship between weight status perceptions and weight loss attempts, strategies, behaviours and outcomes. *Obesity reviews*, 19(3), 347-363.

Lennon, H., Sperrin, M., Badrick, E., & Renehan, A. G. (2016). The obesity paradox in cancer: a review. *Current oncology reports*, 18(9), 56.

Robinson, E., & Oldham, M. (2016). Weight status misperceptions among UK adults: the use of self-reported vs. measured BMI. *BMC obesity*, 3(1), 21.

Vansteelandt, S. (2017). Asking too much of epidemiologic studies: the problem of collider bias and the obesity paradox. *Epidemiology*, 28(5), e47-e49.

Response: In line with our reframing, we have removed this statement, replacing it with the following text:

“However, it was not clear from these studies whether these differences are driven by differences between older and younger people, for example relating to health status. Qualitative research suggests at least some older adults believe carrying extra weight could be protective in times of illness 20, which may mean older people’s weight perceptions are influenced by current perceptions of health status or future health concerns.”

VERSION 2 – REVIEW

REVIEWER	Ashleigh Haynes Cancer Council Victoria, Australia
REVIEW RETURNED	17-Dec-2019

GENERAL COMMENTS	The authors' changes have improved the manuscript greatly, particularly the reframing of the aims to a more general description of weight perception among older adults rather than evidence to support reconsideration of BMI cut offs. I am satisfied that the authors have sufficiently addressed all remaining comments.
--

REVIEWER	Michael Daly Maynooth University
REVIEW RETURNED	18-Dec-2019

GENERAL COMMENTS	The description of the sample and measures and the analytical strategy and results were appropriate in the previous version of the paper. In this version the authors have addressed my comments by providing extensive revisions to ensure the paper is focused squarely on older adults' weight perceptions and sociodemographic factors related to those perceptions. Reference to the obesity paradox is now much less speculative and grounded in both the study findings and a cautious interpretation of the existing literature.
---